# K-SAM: Sharpness-Aware Minimization at the Speed of SGD

## Abstract

Sharpness-Aware Minimization (SAM) has recently emerged as a robust technique for improving the accuracy of deep neural networks. However, SAM incurs a high computational cost in practice, requiring up to twice as much computation as vanilla SGD. The computational challenge posed by SAM arises because each iteration requires both ascent and descent steps and thus double the gradient computations. To address this challenge, we propose to compute gradients in both stages of SAM on only the top-k samples with highest loss. K-SAM is simple and extremely easy-to-implement while providing significant generalization boosts over vanilla SGD at little to no additional cost.

## 1 Introduction

Methods for promoting good generalization are of tremendous value to deep learning practitioners and a point of fascination for deep learning theorists. While machine learning models can easily achieve perfect training accuracy on any dataset given enough parameters, it is unclear when a model will generalize well to test data. Improving the ability of models to extrapolate their knowledge, learned during training, to held out test samples is the key to performing well in the wild.

Recently, there has been a line of work that argues for the geometry of the loss landscape as a major contributor to generalization performance for deep learning models. A number of researchers have argued that flatter minima lead to models that generalize better (Keskar et al., 2016; Xing et al., 2018; Jiang et al., 2019; Smith et al., 2021). Underlying this work is the intuition that small changes in parameters yield perturbations to decision boundaries so that flat minima yield wide-margin decision boundaries (Huang et al., 2020). Motivated by these investigations, Foret et al. (2020) propose an effective algorithm - Sharpness Aware Minimization (SAM) - to optimize models toward flatter minima and better generalization performance. The proposed algorithm entails performing one-step adversarial training in parameter space, finding a loss function minimum that is "flat" in the sense that perturbations to the network parameters in worst-case directions still yield low training loss. This simple concept achieves impressive performance on a wide variety of tasks. For example, Foret et al. (2020) achieved notable improvements on various benchmark vision datasets (e.g., CIFAR-10, ImageNet) by simply swapping out the optimizer. Later, Chen et al. (2021) found that SAM improves sample complexity and performance of vision transformer models so that these transformers are competitive with ResNets even without pre-training. Moreover, further innovations to the SAM setup, such as modifying the radius of the adversarial step to be invariant to parameter re-scaling (Kwon et al., 2021), yield additional improvements to generalization on vision tasks. In addition, Bahri et al. (2021) recently found that SAM not only works in the vision domain, but also improves the performance of language models on GLUE, SuperGLUE, Web Questions, Natural Questions, Trivia QA, and TyDiQA (Wang et al., 2018; 2019; Joshi et al., 2017; Clark et al., 2020).

Despite the simplicity of SAM, the improved performance comes with a steep cost of twice as much compute, given that SAM requires two forward and backward passes for each optimization step: one for the ascent step and another for the descent step. The additional cost may make SAM too expensive for widespread adoption by practitioners, thus motivating studies to decrease the computational cost of SAM. For example, Efficient SAM (Du et al., 2021) decreases the computational cost of SAM by using examples with the largest increase in losses for the descent step. Bahri et al. (2021) randomly select a subset of examples for the ascent step making the ascent step faster. However, both methods still require a full forward and backward pass for either the ascent or descent step on

all samples in the batch, so they are more expensive than vanilla SGD. In comparison to these works, we develop a version of SAM that is as fast as SGD so that practitioners can adopt our variant at no cost.

In this paper, we propose K-SAM, a simple modification to SAM that reduces the computational costs to that of vanilla SGD, while achieving comparable performance to the original SAM optimizer. K-SAM exploits the fact that a small subset of training examples is sufficient for both gradient computation steps (Du et al., 2021; Fan et al., 2017; Bahri et al., 2021), and the examples with largest losses dominate the average gradient over a large batch. To decrease computational cost, we only use the $K$ examples with the largest loss values in the training batch for both gradient computations. When $K$ is chosen properly, our proposed K-SAM can be as fast as vanilla SGD and meanwhile improves generalization by seeking flat minima similarly to SAM.

We empirically verify the effectiveness of our proposed approach across datasets and models. We demonstrate that a small number of samples with high loss produces gradients that are well-aligned with the entire batch loss. Moreover, we show that our proposed method can achieve comparable performance to the original (non-accelerated) SAM for vision tasks, such as image classification on CIFAR-$\{10, 100\}$, and language models on the GLUE benchmark while keeping the training cost roughly as low as vanilla SGD. On the other hand, we observe that on large-scale many-class image classification tasks, such as ImageNet, the average gradient within a batch is broadly distributed, so that a very small number of samples with highest losses are not representative of the batch. This phenomenon makes it hard to simultaneously achieve training efficiency and strong generalization by subsampling. Nonetheless, we show that K-SAM can achieve comparable generalization to SAM with around 65% training cost.

## 2 RELATED WORK

### 2.1 SHARPNESS-AWARE MINIMIZATION

In this section, we briefly introduce how SAM simultaneously minimizes the loss while also decreasing loss sharpness, and we detail why it requires additional gradient steps that in turn double the computational costs during training. Instead of finding parameters yielding low training loss only, SAM attempts to find the parameter vector whose neighborhood possesses uniformly low loss, thus leading to a flat minimum. Formally, SAM achieves this goal by solving the mini-max optimization problem,

$$\min_w L_{\mathcal{S}}^{SAM}(w), \text{ where } L_{\mathcal{S}}^{SAM}(w) = \max_{\|\epsilon\|_2 \leq \rho} L_{\mathcal{S}}(w + \epsilon), \tag{1}$$

where $w$ are the parameters of the neural network, $L$ is the loss function, $\mathcal{S}$ is the training set, and $\epsilon$ is a small perturbation within an $l_2$ ball of norm $\rho$. In order to solve the outer minimization problem, SAM applies a first-order approximation to solve the inner maximization problem,

$$\epsilon^* = \arg\max_{\|\epsilon\|_2 \leq \rho} L_{\mathcal{S}}(w + \epsilon),$$

$$\approx \arg\max_{\|\epsilon\|_2 \leq \rho} L_{\mathcal{S}}(w) + \epsilon^T \nabla_w L_{\mathcal{S}}(w).$$

$$= \rho \nabla_w L_{\mathcal{S}}(w) / \|\nabla_w L_{\mathcal{S}}(w)\|_2. \tag{2}$$

After computing the approximate maximizer $\hat{\epsilon}$, SAM obtains "sharpness aware" gradient for descent:

$$\nabla L_{\mathcal{S}}^{SAM}(w) \approx \nabla_w L_{\mathcal{S}}(w)|_{w+\hat{\epsilon}}. \tag{3}$$

In short, given a base optimizer, i.e., SGD, instead of computing the gradient of the model at the current parameter vector $w$, SAM updates model parameters using the gradient with respect to the perturbed model at $w + \hat{\epsilon}$. Therefore a SAM update step requires two forward and backward passes on each sample in a batch, namely a gradient ascent step to achieve the perturbation $\hat{\epsilon}$ and a gradient descent step to update the current model, which doubles the train time compared to the base optimizer.

### 2.2 EFFICIENT SHARPNESS-AWARE MINIMIZATION

Recently, several works improve the efficiency of SAM while retaining its performance benefits. Bahri et al. (2021) improve the efficiency of SAM by reducing the computational cost of the gradient

ascent step. Instead of computing the ascent gradient on the whole mini-batch, they estimate the gradient using a random subset. This tweak makes the computational cost only about 25% slower than the vanilla SGD training routine when using $1/4$ of the training batch for the ascent gradient, while maintaining performance comparable to SAM. However, reducing the computational cost of only the ascent step cannot make SAM as fast or faster than the base optimizer. In addition, we show in Section 4 that using a random subset of the mini-batch for both gradient computations in SAM has a negative impact on performance, which suggests that we should seek a better selection method than random selection.

Du et al. (2021) propose an efficient SAM (ESAM) algorithm employing two strategies, namely stochastic weight perturbation and sharpness-sensitive data selection. Stochastic weight perturbation updates only part of the weights during the gradient ascent step which offers limited speed-ups. During the descent step, the gradient is only computed using the examples whose loss values increase the most after the parameter perturbation. With these two strategies, ESAM achieves up to 40.3% acceleration compared to SAM. However, since ESAM selects the examples with the largest loss differences, it has to compute the gradient over all samples in the batch for the ascent step and then must similarly perform a forward pass over all samples in the batch for the descent step, which limits the possible speed-ups from this approach.

## 2.3 TOP-K OPTIMIZATION

Top-k optimization has been applied to vanilla SGD (Fan et al., 2017; Kawaguchi & Lu, 2020). Given a training mini-batch, Ordered SGD selects the $K$ examples with the largest losses on which to perform the minimization and computes a subgradient based only on these examples. This work theoretically shows that on convex loss functions, Ordered-SGD is guaranteed to converge sub-linearly to a global optimum and to a critical point with weakly convex losses. In addition, they empirically show that Ordered-SGD can achieve comparable results to vanilla SGD on multiple machine learning models such as SVM and deep neural networks. In this paper, we show that top-k optimization can be effectively applied to both gradient computation steps in SAM and is actually essential to achieving good performance when $K$ is small.

## 3  K-SAM: EFFICIENT SHARPNESS-AWARE MINIMIZATION BY SUBSAMPLING

We now introduce our proposed method, K-SAM, where we select $K_1, K_2$ examples with the largest loss values to estimate the gradients in the ascent and descent steps of a SAM update, respectively. When $K_1$ and $K_2$ are small, the computational complexity of SAM will be vastly reduced since a large proportion of the compute required for neural network training is concentrated in gradient computations.

Recall that in the Section 2.1, given a mini-batch of examples, $\mathcal{B}$, SAM first computes the ascent gradient with respect to the current parameters $w$ to find the perturbation by equation 2. Then, the final gradient descent direction is formed by equation 3. In order to decrease the computational cost, we approximate both gradients by using subsets $\mathcal{M}_{K_1}, \mathcal{M}_{K_2}$ of the mini-batch $\mathcal{B}$ for gradients calculation, where each subset contains training examples with the largest $K_1, K_2$ loss values, respectively. Formally, given the losses, $l = L_{\mathcal{B}}(w)$, of a batch $\mathcal{B}$ and its index set $I = \{1, 2, \cdots, |\mathcal{B}|\}$ of the samples, the subsets can be generated as following,

$$\mathcal{M}_{\{K_1, K_2\}} = \Big\{(x_i, y_i) \in \mathcal{B} : i \in Q_{\{1,2\}}, \quad \text{where} \quad Q_{\{1,2\}} = \underset{Q \subseteq I, |Q| = \{K_1, K_2\}}{\arg\max} \sum_{i \in Q} l_i\Big\}.$$

Given these subsets, we can efficiently obtain the ascent gradients, and then achieve adversarial perturbation in parameter space by

$$\hat{\epsilon} = \rho \nabla_w L_{\mathcal{M}_{K_1}}(w) / \|\nabla_w L_{\mathcal{M}_{K_1}}(w)\|_2.$$

Finally, we get the actual descent gradient $\hat{g}_{SAM}$ of SAM via

$$\hat{g}_{SAM} = \nabla_w L_{\mathcal{M}_{K_2}}(w)|_{w+\hat{\epsilon}}.$$

The detailed formulation can be found in Algorithm 1.

---

**Algorithm 1** K-SAM

---

1: **Input:** training set $\mathcal{S}$, loss function $L$, batch size $b$, ascent subset size $K_1$, descent subset size $K_2$, neighborhood size $\rho$, model parameter $w$, learning rate $\eta$, iterations $T$.
2: **for** $i = 1$ **to** $T$ **do**
3:     Sample mini-batch $\mathcal{B} \in \mathcal{S}$ with size $b$.
4:     Get the loss values for the mini-batch: $l_\mathcal{B} = L_\mathcal{B}(w)$.
5:     Generate ascent batch $\mathcal{M}_{K1}$ and descent batch $\mathcal{M}_{K2}$ with largest $K_1$, $K_2$ losses respectively.
6:     Compute perturbation for ascent step : $\hat{\epsilon} = \rho \frac{\nabla_w L_{\mathcal{M}_{K_1}}(w)}{\|\nabla_w L_{\mathcal{M}_{K_1}}(w)\|_2}$.
7:     Compute descent gradient of SAM: $\hat{g}_{SAM} = \nabla_w L_{\mathcal{M}_{K_2}}(w)|_{w+\hat{\epsilon}}$.
8:     Update parameters: $w = w - \eta * \hat{g}_{SAM}$.

---

### 3.1 GRADIENTS OF K-SAM ARE ACCURATE AND EFFICIENT APPROXIMATIONS

K-SAM is motivated by the observation that gradients of both the ascent and descent steps in SAM can be accurately approximated by gradients calculated from the subsets with the largest $k$ losses (top-k losses). The intuition underlying this phenomenon is that the gradient on a mini-batch of data is an average of the individual per-sample gradients, and this average is dominated by a small number of terms with large gradient magnitudes. Since we can cheaply select samples with high loss, these will serve as effective surrogates. We will investigate both key components, ascent and descent gradient approximation, in detail to validate this hypothesis.

#### 3.1.1 GRADIENT APPROXIMATION IN ASCENT STEPS

We first analyze how accurately the gradients, $\nabla_w L_{\mathcal{M}_{K_1}}(w)$, of the subset with top-k losses approximate the gradients, $\nabla_w L_\mathcal{B}(w)$, in ascent steps of SAM, by measuring the cosine similarity of the two gradients during training. We train a WideResNet-16-8 (Zagoruyko & Komodakis, 2016) with SAM and batch size 128 on CIFAR-10. In the left plot of Figure 1, we observe that the alignments between the top-k gradients and those computed on the full mini-batch in the ascent steps are very high, especially at the beginning and the middle of the training. For example, we see a cosine similarity of around 0.9 when $k = 64$. In addition, compared to gradients computed on a randomly chosen subset of $k$ examples (random-$k$) (Bahri et al., 2021), we find that top-k gradients approximate the gradients of the full mini-batch in the ascent steps far better. When we use the same number of examples $k$, top-32 is almost twice as well aligned with the full batch ascent gradients as random-32. Even when we only use 16 examples with the largest losses, we are still able to estimate the ascent gradients up to 50% more accurately compared to random-32. This difference between the two approximations is especially prominent during the central parts of training. On the other hand, if we want to estimate the full mini-batch gradients as accurately as top-16, we would have to use 4 times as many examples with random selection, as visualized in the middle plot of Figure 1. In summary, top-k is much more efficient and accurate for estimating gradients in the ascent steps compared to random subsets.

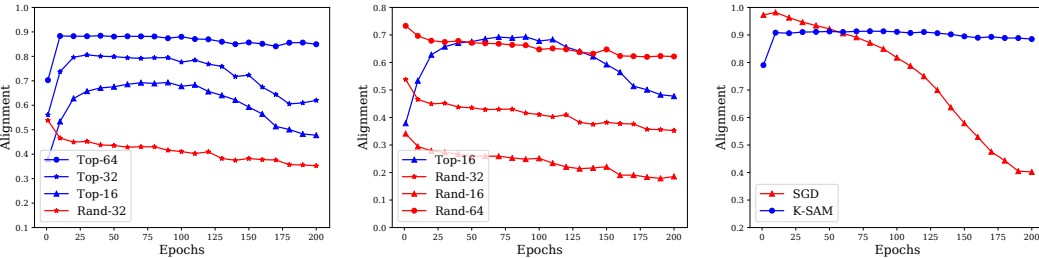

Figure 1: (Left) Gradient alignment between full mini-batch and subsets of largest $K_1 = \{16, 32, 64\}$ losses. (Middle) Gradient Alignment between full mini-batch and subsets of random $K_1 = \{16, 32, 64\}$ losses. (Right) Gradient Alignment between SAM updates and updates of SGD and K-SAM.

Table 1: Using a subset to update the model with SGD on CIFAR-10 and CIFAR-100. We find that using top-k subsets yields similar performance to SGD, while using random-k subsets exhibits a clear performance drop.

| Model | Accuracy(%) | |
|---|---|---|
| ResNet-18 | CIFAR-10 | CIFAR-100 |
| SGD | $96.29 \pm 0.09$ | $79.08 \pm 0.18$ |
| top-64 | $96.26 \pm 0.14$ | $79.04 \pm 0.13$ |
| random-64 | $95.92 \pm 0.06$ | $78.53 \pm 0.20$ |
| Wide-28-10 | CIFAR-10 | CIFAR-100 |
| SGD | $96.99 \pm 0.09$ | $82.05 \pm 0.15$ |
| top-64 | $97.02 \pm 0.14$ | $82.29 \pm 0.22$ |
| random-64 | $96.66 \pm 0.09$ | $82.31 \pm 0.17$ |

### 3.1.2 Gradient Approximation in Descent Steps

Now that we can reasonably approximate the gradients in ascent steps with fewer samples, we analyze whether or not we can estimate the gradients in descent steps accurately with a similar strategy. We show in Table 1 that comparable performance can be achieved when we compute the top-k gradients and perform descent using this approximation to vanilla SGD (rather than SAM) for a range of settings. Using top-k descent gradients is thus a sensible strategy, even for the updates of deep neural networks, an extension of theoretical investigations of convex optimization in Fan et al. (2017) and Kawaguchi & Lu (2020). We conduct these experiments on both CIFAR-10 and CIFAR-100 datasets with ResNet-18 and WideResNet-28-10 models. In the forward pass, we feed the full mini-batch of size 128 into the network. In the backward pass, we compute the gradients with 1) the full training batch (SGD), 2) half of the training batch with the largest loss values (top-64) and 3) half of the training batch with random sampling (random-64). As shown in Table 1, training with only the 64 samples with the largest losses yields performance comparable, and sometimes even superior, to vanilla SGD on both network architectures and datasets. However, when we do gradient descent based on a random subset of 64 examples, we see a clear accuracy drop in most experiments. The accuracy drop suggests that using top-k subsets is a better approximation of the descent gradients of the full mini-batch.

In addition, we measure the cosine similarity between the gradients in descent steps of K-SAM and SAM, where the same ascent gradients are used for both methods. In the right plot of Figure 1, we observe that throughout training, K-SAM's estimated descent gradients are very similar to SAM's descent gradients with a cosine similarity of 0.9. We also show the similarity between the gradients in descent steps of SGD and SAM for comparison, where SGD's descent gradient is quite similar to SAM in the first few epochs but drops dramatically late in training. This analysis reveals that the K-SAM's approximation of SAM systemically points in the intended direction and toward flatter minima.

### 3.2 Efficiency Gains from K-SAM

We implement K-SAM in the following way: we first compute a forward pass on the full mini-batch, and then we perform forward and backward propagation on both subsets $\mathcal{M}_{K_1}$ and $\mathcal{M}_{K_2}$. Theoretically, if we want K-SAM to produce exactly the same computational cost as vanilla SGD (ignoring the cost of loading data into memory), we have to choose $K_1, K_2$ such that $K_1 + K_2 = B/4$, where $B$ is the batch size. However, In practice, due to the fact that we do not need to compute the gradient for the first forward pass, we can benefit from significant speed-up and memory savings, as analyzed in Table 2. In addition, this first forward pass can be further accelerated by computing it with lower precision, given that only a ranking of the output values is necessary. For example, on a conventional `NVIDIA 2080ti` card, a forward pass in mixed lower precision and inference mode (26ms) can be twice as fast as an (already fast) normal forward pass (60ms) as described in Table 2. As a result, we do not need to select $K_1$ and $K_2$ as small as one-fourth of the batch size. Indeed, we empirically find that when we set $K_1 = B/8$ and $K_2 = B/2$, the proposed K-SAM is nearly as fast as vanilla SGD (See 4.2). In addition, we can further increase the efficiency of K-SAM by gradually

Table 2: Time comparison between a full forward-backward pass, a regular forward pass, a forward pass in inference mode with mixed precision, and a forward pass on a subset of the batch. The model is a ResNet-50 and batch size is $48$. All numbers are based on the average of 250 measurements on a `NVIDIA 2080ti` card each after CUDA synchronization. Note that larger batch sizes have more potential for efficiency gains on smaller reductions - here, we focus on a conventional device and scenario where we find that the $16-$fold reduction only halves the runtime compared to the $4-$fold reduction, due to GPU undersaturation - however, the proposed approach still succeeds in reaching SGD runtimes.

| Operation | Time (ms) |
|---|---|
| Regular forward pass | 56.81 |
| Forward and backward pass | 189.92 |
| Mixed-precision forward pass | 25.93 |
| 16-fold reduced forward-backward pass | 26.16 |
| 4-fold reduced forward-backward pass | 55.31 |

decreasing $K_2$ during training, thus leading to an algorithm that often runs even faster than vanilla SGD on conventional hardware. We include PyTorch-like pseudo-code in Appendix 2, where we summarize all steps discussed so far in detail.

## 4 EXPERIMENTS

This section contains experimental evaluations of the proposed K-SAM optimization scheme. We run all experiments using conventional widely available GPUs and evaluate realistic PyTorch code without additional customized modifications for speed. We note that although we could make the first forward pass faster by using mixed-precision arithmetic as described in Section 3.2, we keep the precision unchanged for all experiments for fair comparisons. We evaluate the effectiveness of K-SAM on image classification tasks CIFAR-10, CIFAR-100, and ImageNet, and on the General Language Understanding Evaluation (GLUE) benchmark. In addition, ablation studies are conducted to further show the robust effectiveness of our proposed method under hyperparameter selections.

### 4.1 BENCHMARK EVALUATIONS

**CIFAR-10 and CIFAR-100.** We first show that K-SAM can achieve comparable performance to SAM on both CIFAR-10 and CIFAR-100 but at a comparable runtime to vanilla SGD. We evaluate the proposed K-SAM for three different network architectures: ResNet-18 (He et al., 2016), WideResNet-28-10 (Zagoruyko & Komodakis, 2016) and PyramidNet-110 (Han et al., 2017). Unless otherwise stated, we use the Inception-style (Szegedy et al., 2016) image pre-processing and Cutout (DeVries & Taylor, 2017) as default data augmentations. For each experiment, we evaluate over 5 trials and report the mean as well as confidence intervals with width of one standard error. In Table 3, we compare test classification accuracy as well as training time among optimizers: SGD, SAM, ESAM, and our proposed K-SAM. Within the same architecture, we use the same hyper-parameters for all optimizers. For ResNet-18 and WideResNet-28-10, we train 200 epochs with a cosine learning rate scheduler and peak learning rate $0.05$. For PyramidNet-110, we train for 300 epochs with cosine learning rate scheduler and peak learning rate $0.1$. We use a perturbation radius of $\rho = 0.05$ for ResNet-18 and $\rho = 0.1$ for WideResNet-28-10 and PyramidNet-110. Unless otherwise stated, we train all models with the same batch size 128 on a single GPU.

Starting from the default batch size of 128, we set $K_1 = 16$, $K_2 = 64$ for K-SAM. To accelerate training, we further decrease $K_2$ by half midway through training, denoted by K-SAM$^*$. As can be seen in Table 3, the test accuracy of K-SAM is comparable to SAM while the training speed remains almost at the level of SGD. On average, without K-SAM$^*$, we add $\sim 6\%$ runtime on top of vanilla SGD, which is significantly less than ESAM which is $\sim 30\%$ slower than SGD. When we switch to K-SAM$^*$, we can achieve test accuracy on par with SAM at even faster speeds than SGD in 5 out of 6 experiments. For example, for WideResNet-28-10 trained on CIFAR-10, K-SAM$^*$ is $\sim 10\%$ faster than SGD while obtaining comparable accuracy to SAM (within $0.04\%$). In addition, since

Table 3: Classification accuracy and training efficiency on the CIFAR-10 and CIFAR-100 datasets. The efficiency is evaluated by total training time in hours and percentage of GPU memory allocation during training. Classification accuracy mean is calculated from 5 runs. ∗ denotes the faster version of proposed K-SAM, where $K_2$ gradually decays during training. K-SAM can achieve comparable performance as SAM while adding little computational cost on the base optimizer, i.e., SGD.

| Method | Backbone | CIFAR-10 | | | CIFAR-100 | | |
|---|---|---|---|---|---|---|---|
| | | Accuracy (%) | Train Time(h) | Memory(%) | Accuracy(%) | Train Time(h) | Memory(%) |
| SGD | ResNet-18 | $96.29 \pm 0.09$ | 1.07 | 21.50 | $79.08 \pm 0.18$ | 1.28 | 21.20 |
| SAM | ResNet-18 | $96.69 \pm 0.13$ | 2.03 | 20.85 | $80.08 \pm 0.19$ | 2.04 | 20.85 |
| ESAM | ResNet-18 | $96.55 \pm 0.13$ | 1.48 | 21.92 | $79.34 \pm 0.24$ | 1.59 | 20.75 |
| K-SAM | ResNet-18 | $96.54 \pm 0.14$ | 1.13 | 17.48 | $79.24 \pm 0.29$ | 1.18 | 17.48 |
| K-SAM* | ResNet-18 | $96.47 \pm 0.05$ | 1.05 | 17.48 | $79.00 \pm 0.16$ | 1.08 | 17.48 |
| SGD | Wide-28-10 | $96.99 \pm 0.09$ | 6.19 | 43.52 | $82.05 \pm 0.15$ | 5.96 | 45.05 |
| SAM | Wide-28-10 | $97.45 \pm 0.07$ | 12.02 | 51.45 | $84.16 \pm 0.16$ | 11.44 | 47.90 |
| ESAM | Wide-28-10 | $97.29 \pm 0.06$ | 8.15 | 38.25 | $84.21 \pm 0.14$ | 7.98 | 38.28 |
| K-SAM | Wide-28-10 | $97.45 \pm 0.07$ | 6.28 | 36.48 | $84.01 \pm 0.29$ | 6.21 | 36.50 |
| K-SAM* | Wide-28-10 | $97.41 \pm 0.05$ | 5.55 | 35.08 | $84.05 \pm 0.19$ | 5.60 | 35.08 |
| SGD | PyramidNet-110 | $97.07 \pm 0.11$ | 11.80 | 59.66 | $83.45 \pm 0.24$ | 11.22 | 64.01 |
| SAM | PyramidNet-110 | $97.82 \pm 0.11$ | 22.50 | 70.56 | $85.95 \pm 0.21$ | 23.18 | 70.57 |
| ESAM | PyramidNet-110 | $97.71 \pm 0.07$ | 19.65 | 64.68 | $85.41 \pm 0.32$ | 19.78 | 65.07 |
| K-SAM | PyramidNet-110 | $97.60 \pm 0.06$ | 12.56 | 41.20 | $85.38 \pm 0.18$ | 12.82 | 41.25 |
| K-SAM* | PyramidNet-110 | $97.62 \pm 0.10$ | 11.53 | 40.03 | $84.60 \pm 0.22$ | 11.90 | 40.07 |

we only compute forward passes on the full mini-batch and never compute a full backward pass, which is restricted merely to the subsets $\mathcal{M}_{K_1}, \mathcal{M}_{K_2}$, the memory required by K-SAM is actually smaller than both SGD and SAM. Table 3 shows the memory allocation on a single GPU during training. On average, K-SAM saves around 20% of GPU memory compared to vanilla SGD.

**ImageNet** To verify our proposed method on ImageNet, we train ResNet-50 models for 90 epochs with a cosine annealing learning rate scheduler and peak learning rate 0.05. For both SAM and K-SAM, we set perturbation radius $\rho = 0.05$ and train all the models with the same batch size 512 on 8 GPUs in parallel (64 images per GPU). We vary the size of both subsets $\mathcal{M}_{K_1}$ and $\mathcal{M}_{K_2}$ from $\{B/8, B/4, B/2, B\}$, where $B$ is the batch size. Contrary to our observations on CIFAR-10 and CIFAR-100, here we observe a clear trade-off between the size $K_1, K_2$ of the subsets and the test accuracy. In Appendix A (see Table 7 and Table 8), we can see that the test accuracy drops severely when we have fewer samples to estimate the gradients, especially for the descent steps. To understand this phenomenon, we measure the gradient alignments for ImageNet models as well. In Figure 2, we show that in general, ImageNet models have worse alignment between gradients of SAM and K-SAM, especially when the size of subsets is small, i.e., $K_1 = B/8$.

Although we do not achieve the same speed as SGD while enjoying the same generalization power as SAM, we show in Table 4 that we can achieve even better results than SAM yet with only 80% of its training cost. In addition, we can achieve comparable performance to SAM (0.1% accuracy drop), but with only 65% of its training cost.

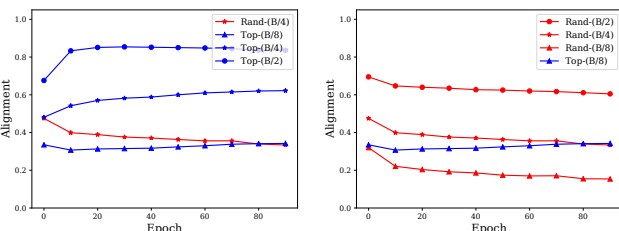

Figure 2: Gradient alignments between full mini-batch and subsets of {Top, Random}-$\{B/2, B/4, B/8\}$ losses for ImageNet, where $B$ is the batch size. In general, the alignments are small, especially when $K = B/8$. Again, gradients calculated on samples with top-k losses align better compared to random sampling.

Table 4: Performance of SGD, SAM and K-SAM on ImageNet. K-SAM achieves comparable testing accuracy to SAM, while reducing the computational cost greatly.

| ResNet-50 | Accuracy(%) | Images/s | ResNet-50 | Accuracy(%) | Images/s |
|---|---|---|---|---|---|
| SGD | 75.98 | 1664.57 | K-256/360 | 77.08 | 1192.67 |
| SAM | 77.19 | 775.80 | K-256/512 | 77.28 | 935.86 |

**GLUE benchmark** In this section, we test the effectiveness of K-SAM on the GLUE benchmark. We fine-tune the BERT-BASE pre-trained model (Devlin et al., 2018) on the 8 NLP tasks with a batch size of 56 for 250k steps. We use perturbation radius $\rho = 0.02$ for all experiments. For our proposed K-SAM, we set $K_1$ and $K_2$ to 8 and 32, respectively. We find that on most tasks, SAM improves the performance of the model comparing to SGD, and K-SAM yields similar improvement to SAM. When excluding the **rte** dataset, a special case in which SGD significantly outperforms SAM and its variants, SAM and K-SAM improve the performance of SGD by $0.25\%$ on average (see Table 5). Notably, our observations concerning the **rte** dataset are consistent with observations in (Bahri et al., 2021), where the performances of both SAM and K-SAM are very poor, and the performance gap dominates the average over datasets.

Table 5: Performance of SAM and K-SAM on the GLUE benchmark. Comparing to the base optimizer SGD, K-SAM yields similar improvement to SAM on most tasks, except on the **rte** dataset.

| | Overall | Overall (exclude rte) | cola | sst2 | mrpc | | qqp | | stsb | | mnli | qnli | rte |
|---|---|---|---|---|---|---|---|---|---|---|---|---|---|
| SGD | **83.51** | 85.12 | 61.40 | 92.20 | **88.24** | 91.31 | 91.27 | 88.20 | 88.12 | 87.85 | 83.62 | 91.14 | **72.20** |
| SAM | 83.18 | 85.37 | 61.36 | **92.66** | 87.50 | **91.75** | 91.53 | **88.60** | 88.68 | 88.29 | **84.08** | 91.32 | 67.87 |
| K-SAM | 83.15 | **85.38** | **63.06** | 92.32 | 86.03 | 90.36 | 91.21 | 88.19 | **89.80** | **89.51** | 83.35 | **91.38** | 67.51 |

## 4.2 EFFECTIVENESS OF K-SAM ON DIFFERENT SETTINGS

If selecting samples with high loss is to be useful, the gradients of these samples must be more representative of the full batch than a random subset. And moreover, we need to know exactly how big our selected subsets must be. We now answer these questions by performing a series of ablations, and we additionally put K-SAM to the test in the distributed training to show that the same efficiency benefits can be achieved in such a setting.

### 4.2.1 K-SAM PERFORMS BETTER THAN RANDOMLY SAMPLED SUBSETS

Bahri et al. (2021) suggest that SAM can be made more efficient by randomly sampling $1/4$ of the batch for the computation of the model perturbation. To compare this variant with K-SAM, we train ResNet-18 models on both CIFAR-10 and CIFAR-100 with both optimizers and batch size 128. We evaluate both methods across various $K_1$ and $K_2$. We see in Table 6 that random subsets can achieve comparable results to K-SAM only when either $K_1$ or $K_2$ is large. However, when we use smaller $K_1, K_2$, that is one-eighth and half of the batch size, accuracy noticeably drops when using the random subsets for computing ascent directions, which prevents further subsampling acceleration for descent steps.

### 4.2.2 DIFFERENT SIZES OF TOP-K SUBSETS

The size of both subsets $K_1$ and $K_2$ offers a trade-off between training speed and the fidelity with which we approximate SAM. To evaluate the effect of $K_1, K_2$, we train K-SAM with both ResNet-18 and WideResNet-28-10 on CIFAR-10 and CIFAR-100 with various $K_1, K_2$ from set $\{16, 64, 128\}$, where batch size equals to 128. We provide the results in Appendix C, where Table 9 shows the performance and efficiency across combinations of $K_1, K_2$. Interestingly, we usually achieve the highest classification accuracy (even higher than SAM on WideResNet-28-10) when $K_1 = K_2 = 64$, which offers a $\sim 30\%$ speed-up compared to SAM.

Table 6: Comparison between sampling methods for subsets $\mathcal{M}_{K_1}$ used for parameter perturbations (ascent). "Rand-$K_1/K_2$" denotes random sampling and "K-$K_1/K_2$" denotes K-SAM. K-SAM outperforms random sampling when both $K_1$ and $K_2$ are small.

| ResNet-18 | CIFAR-10 | | CIFAR-100 | |
|---|---|---|---|---|
| | Accuracy(%) | Time(h) | Accuracy(%) | Time(h) |
| SGD | $96.29 \pm 0.09$ | 1.07 | $79.08 \pm 0.18$ | 1.28 |
| Rand-16/128 | $96.41 \pm 0.09$ | 1.35 | $79.35 \pm 0.22$ | 1.28 |
| Rand-16/64 | $96.22 \pm 0.15$ | 1.15 | $79.01 \pm 0.31$ | 1.15 |
| Rand-32/128 | $96.59 \pm 0.07$ | 1.39 | $79.53 \pm 0.38$ | 1.28 |
| Rand-32/64 | $96.34 \pm 0.15$ | 1.19 | $79.49 \pm 0.24$ | 1.16 |
| K-16/128 | $96.58 \pm 0.12$ | 1.53 | $79.17 \pm 0.32$ | 1.62 |
| K-16/64 | $96.54 \pm 0.14$ | 1.13 | $79.24 \pm 0.29$ | 1.18 |
| K-32/128 | $96.61 \pm 0.09$ | 1.64 | $79.55 \pm 0.20$ | 1.64 |
| K-32/64 | $96.61 \pm 0.03$ | 1.17 | $79.45 \pm 0.25$ | 1.17 |

### 4.2.3 DISTRIBUTED LEARNING

In this section, we verify that the effectiveness and efficiency of K-SAM will hold in the distributed setting as well when the batch size is larger and multiple GPUs are used. In the distributed setting, the ascent gradients are calculated by the training examples on each card as described in Foret et al. (2020). The gradients are only aggregated after the descent gradients are calculated. We train the WideResNet-16-8 model with different optimizers, using batch size 512 on 4 GPUs. The results are provided in Appendix C (Table 10). We find that K-SAM can still achieve comparable performance and great efficiency improvements in this distributed setting. For some combination of $K_1$ and $K_2$, K-SAM even outperforms the baseline SAM but with a significantly lower computational cost. On CIFAR-100, for example, we can obtain performance greater than that of SAM within a minute of the runtime of vanilla SGD.

## 5 CONCLUSION

In this paper, we explore a mechanism for improving the training efficiency of Sharpness-Aware Minimization (SAM). We find that a simple top-k selection strategy for both ascent and descent steps is a succinct and effective way for accelerating SAM training. We validate the underlying intuition that gradients calculated on training examples with the largest several losses are good approximations for the gradients in both steps of SAM by measuring the cosine similarity between the gradients in SAM and their approximations. Compared to random subsampling, we show that gradients calculated on top-k samples align better. In addition, we empirically evaluate our method, K-SAM, on multiple vision and language benchmarks, and show that our method can achieve comparable performance to SAM while reducing the computational cost significantly. Although K-SAM can accelerate the sharpness-aware training, it still needs an additional forward pass on the full minibatch. How to avoid this pass or even avoid the ascent step in SAM will be an interesting future direction.

### REPRODUCIBILITY STATEMENT

We ran all the experiments with specific random seeds and multiple runs. We provide the source code in the supplementary material, and all the results are reproducible.

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

# K-SAM: Sharpness-Aware Minimization at the Speed pf SGD (Appendix)

## A   TRADE-OFF BETWEEN THE SIZE OF SUBSETS AND K-SAM PERFORMANCE ON IMAGENET

We provide results for different combinations of $K_1$ and $K_2$ for K-SAM on ImageNet with model ResNet-50. We observe a clear trade-offs between the size of subsets and the testing accuracy, where when we have smaller $K$, especially $K_2$, the performance on ImageNet will drop significantly. In addition, we show that this happens to SGD as well, where the accuracy drops more than 1% when we only backpropagate through half of the mini-batches.

Table 7: K-SAM has a clear trade-offs on ImageNet between the size of the subsets and testing accuracy.

| $K_1/K_2$ | 256 | 360 | 512 |
|---|---|---|---|
| 64 | 75.37 | 76.43 | 76.55 |
| 128 | 75.99 | 76.81 | 76.73 |
| 256 | 76.08 | 77.09 | 77.29 |
| 512 | 76.34 | 76.90 | 77.19 |

Table 8: Performance drops significantly on ImageNet if $K_2$ is small when applied to vanilla SGD.

| $K_2$ | 256 | 360 | 512 |
|---|---|---|---|
| SGD | 74.866 | 75.516 | 75.978 |

# B PYTORCH-LIKE PSEUDO CODE FOR K-SAM

The PyTorch-Like pseudo code is proved in Alg 2.

---

**Algorithm 2** PyTorch-like implementation for K-SAM

```
# Pytorch-like training loop

for inputs, targets in data_loader:

    with torch.no_grad():

        outputs = model(inputs)

        loss = criterion(outputs, targets)

        _, idx_k1 = torch.topk(loss, K₁) # Get M_{K₁}

        _, idx_k2 = torch.topk(loss, K₂) # Get M_{K₂}

    # Get the model perturbation

    outputs_k = model(inputs[idx_k1])
    loss_k1 = criterion(outputs_k, targets[idx_k1])
    loss_k1.mean().backward()
    optimizer.first_step()
    # Get the SAM updates

    outputs_k = model(inputs[idx_k2])
    loss_k2 = criterion(outputs_k, targets[idx_k2])
    loss_k2.mean().backward()
    optimizer.second_step()
```

---

# C ABLATION STUDIES

**Different Subset Sizes** Contrary to ImageNet, K-SAM consistently outperforms SGD with different $K_1, K_2$ on CIFAR-10 and CIFAR-100. In addition, the best performance is usually achieved when $K_1 = K_2 = B/2$, where $B$ is the batch size, which is almost as good as or even better than the original SAM.

Table 9: Classification accuracy and total training time for K-SAM across combinations of $K_1, K_2$. Best accuracy is usually achieved when both of $K_1, K_2$ equal to half of the batch size.

| | CIFAR-10 | | CIFAR-100 | |
|---|---|---|---|---|
| ResNet-18 | Accuracy(%) | Time(h) | Accuracy(%) | Time(h) |
| SGD | $96.29 \pm 0.09$ | 1.07 | $79.08 \pm 0.18$ | 1.28 |
| K-128/64 | $96.59 \pm 0.03$ | 1.64 | $79.83 \pm 0.16$ | 1.77 |
| K-64/64 | $96.64 \pm 0.04$ | 1.36 | $79.75 \pm 0.22$ | 1.26 |
| K-64/128 | $96.63 \pm 0.08$ | 1.74 | $79.86 \pm 0.28$ | 1.84 |
| K-16/64 | $96.54 \pm 0.14$ | 1.13 | $79.24 \pm 0.29$ | 1.18 |
| Wide-28-10 | Accuracy(%) | Time(h) | Accuracy(%) | Time(h) |
| SGD | $96.91 \pm 0.07$ | 6.19 | $82.05 \pm 0.15$ | 5.96 |
| K-128/64 | $97.46 \pm 0.10$ | 10.84 | $84.39 \pm 0.20$ | 10.61 |
| K-64/64 | $97.46 \pm 0.04$ | 8.13 | $84.52 \pm 0.15$ | 8.36 |
| K-64/128 | $97.45 \pm 0.05$ | 10.64 | $84.17 \pm 0.15$ | 10.79 |
| K-16/64 | $97.45 \pm 0.07$ | 6.28 | $84.01 \pm 0.29$ | 6.21 |

**Effectiveness and Efficiency under distributed learning** In Table 10, we show that in the distributed setting, where vanilla SGD and SAM will be faster, we can still achieve the same efficiency improvements by K-SAM. In addition, K-SAM will achieve comparable results to SAM as well.

Table 10: Top-k SAM in the distributed larger batch setting. In this table, we observe that results in smaller batch setting extend to larger batch sizes and distributed computation. Experiments in this table use a batch size of 512 and $\rho = 0.02$ to train a WideResNet-16-8.

| Wide-16-8 | CIFAR-10 | | CIFAR-100 | |
|---|---|---|---|---|
| | Accuracy(%) | Time (minutes) | Accuracy(%) | Time(minutes) |
| SGD | $96.50 \pm 0.10$ | **47** | $80.01 \pm 0.09$ | 55 |
| SAM | $96.70 \pm 0.13$ | 93 | $80.35 \pm 0.23$ | 91 |
| K-64/512 | **96.79** $\pm 0.07$ | 62 | $80.17 \pm 0.28$ | 60 |
| K-64/384 | $96.53 \pm 0.10$ | 55 | **80.45** $\pm 0.28$ | 56 |
| K-64/256 | $96.28 \pm 0.12$ | **47** | $80.30 \pm 0.08$ | **46** |

