# OpenReview forum: "K-SAM: Sharpness-Aware Minimization at the Speed of SGD"
_ICLR.cc/2023/Conference — Submitted to ICLR 2023_

### Official Review · Reviewer_T2UG · 2022-10-17

**Confidence:** 4
**Correctness:** 3
**Technical Novelty And Significance:** 2
**Empirical Novelty And Significance:** 2
**Recommendation:** 3

**Clarity, Quality, Novelty And Reproducibility:**

### Clarity

As stated above, the paper is well-written and adequately positioned in the context of related work. The proposed algorithm is described in full detail.

### Quality

The experimental comparison is very generally very well done, but I think one very important baseline is missing (see weakness 2).

### Originality

This is a rather incremental paper, applying the top-k trick, known to work on SGD, to SAM.

**Strength And Weaknesses:**

### Strengths

1) The paper takes a highly relevant technique, SAM, and improves its computational efficiency. In my opinion that is commendable work with a direct positive impact on practitioners.

2) The paper is well-written. It does a good job of positioning itself in the context of existing work and summarizing the work it builds upon. Overall, the claims are scoped and phrased adequately.

3) The experimental comparison is well done. The selected benchmark problems offer a good amount of variety. The most relevant ablation/sensitivity studies (random examples instead of top-$k$, effect of $K_1$ and $K_2$) are included.


### Weaknesses

1) The paper is of limited originality. It combines the known optimization method SAM with the known (from vanilla SGD) speed-up method of selecting the examples with the top-k losses. Personally, I am fine with such a rather incremental paper, but it requires excellent execution.

2) My main criticism is that I sorely miss one baseline to compare against in the experiments: top-$K$-SGD (e.g. with $K=K_1+K_2$). The paper evaluates methods with regards to their trade-off between generalization and computational cost. For SAM, it uses the "top-$k$ trick", for SGD it does not. In my view, that is an incomplete picture. Assuming that top-$K$-SGD can maintain the generalization performance of SGD for reasonable values of $K$ it would naturally be the fastest method. This would make the claim of providing "significant generalization boosts over [...] SGD at little to no additional costs" rather vacuous. (Importantly, I don't think that would reduce the value of the paper. It is still worthwhile investigating the application of the top-k trick to SAM. But I _do_ think that this baseline is needed to give the full picture.)

3) The proposed methods computes the ranking based on the losses computed for the _ascent_ step and then uses that ranking for both the ascent and the descent step. In my view, the most straight-forward way of transferring the top-k trick from vanilla SGD to SAM would be to select the top-k examples _separately_ for the ascent step and the descent step. Since these loss values would be obtained at $w$ and $w+\varepsilon_\ast$, respectively, this could lead to slightly different results. Using the ranking from the ascent step might be a totally adequate strategy, but I think the first option should be mentioned/discussed in the paper and it would be nice to see an ablation study on that.


### Update after Rebuttal

Thank you for your response. I see now that Table 8 in the appendix contains numbers for top-K-SGD on ImageNet. (However, that is a _single_ experiment.) If we compare top-K-SGD with K-SAM such that $K=K_1 + K_2$ for (roughly) equal computational cost, the test performance achieved by the two methods is actually quite close. In any case, I think this is an absolutely crucial baseline to include for all experiments and contrast the results for K-SAM with. Therefore, unfortunately this doesn't resolve my concern outlined in the original review and I will keep my recommendation as is.

**Summary Of The Paper:**

The paper investigates the application of the "top-k trick" known from SGD, where only the $k$ examples with the highest loss values are used in backward pass, to the sharpness-aware minimization (SAM) method. It shows via extensive experiments that this can reduce the computational cost of SAM to the level of vanilla SGD while maintaining the generalization-boosting effects of SAM.

**Summary Of The Review:**

This is a well-written paper providing a small but significant improvement to a relevant technique such as SAM and demonstrates that improvement in extensive experiments. In terms of originality and scientific value, this is certainly a very incremental paper. That, together with the missing baseline of top-$K$-SGD (weakness 2), puts this paper below the bar for a top conference such as ICLR.

---

> ### Author Response · Authors · 2022-11-17
> **Thanks for the review**
>
> 1.  **My main criticism is that I sorely miss one baseline to compare against in the experiments: top-K-SGD (e.g. with K=K1+K2).**
>
>     Thank you for pointing this out. We actually have the result for K=32, which is half of the batch size. We show that SGD suffers from an accuracy drop in such cases. It might be interesting to tweak this K for different values to see how it compares to vanilla SGD.
>
> 2. **In my view, the most straight-forward way of transferring the top-k trick from vanilla SGD to SAM would be to select the top-k examples separately for the ascent step and the descent step.**
>
>    Thanks for the suggestion, indeed, selecting top-k examples separately for the ascent step and the descent step is a more straightforward estimation. However, in order to get both steps separately, we have to do two forward passes on the whole batch which will be definitely more expensive than vanilla SGD.  On the contrary, our method shows a way that SAM can be as cheap as SGD while enjoying accuracy improvement.

---

> > ### Comment · Reviewer_T2UG · 2022-12-14
> > **Response**
> >
> > Thank you for your response. I see now that Table 8 in the appendix contains numbers for top-K-SGD on ImageNet. (However, that is a single experiment.) If we compare top-K-SGD with K-SAM such that $K=K_1 + K_2$  for (roughly) equal computational cost, the test performance achieved by the two methods is actually quite close. In any case, I think this is an absolutely crucial baseline to include for all experiments and contrast the results for K-SAM with. Therefore, unfortunately this doesn't resolve my concern outlined in the original review and I will keep my recommendation as is.

---

### Official Review · Reviewer_Y4Jo · 2022-10-20

**Confidence:** 4
**Correctness:** 3
**Technical Novelty And Significance:** 2
**Empirical Novelty And Significance:** 2
**Recommendation:** 3

**Clarity, Quality, Novelty And Reproducibility:**

### Clarity
The writing is generally good. But it's not clear to me how to obtain the K1 and K2 example. Additionally, there are too many similar tables in the paper, the authors should consider merging / reorganizing them.

### Novelty
The idea is simple and straightforward, and selecting the top-K hardest examples has a long history.

### Reproducibility
Code is provided so I assume that the reproducibility is good.

**Strength And Weaknesses:**

### Strengths
- Motivation of this paper is clear as SAM usually doubles the computational cost of SGD.

### Weaknesses
- How do you obtain the K1 and K2 example specifically? Do you just randomly choose from the K1+K2 examples that have the largest loss? Will their order affect the result? The authors should elaborate more on the algorithm.
- To me, it's surprising to see in Table 1 that just using top-K examples in SGD bring little performance drop. One possible reason is that the   batch size 128 used here is pretty small. Can the authors also provide analysis with a larger batch size.
- The reported training time shows that K-SAM can achieve speedup compared to SGD, while on ImageNet it's much slower. Can the authors provide metrics like FLOPs, which is not impacted by the hardware.
- It could be better if the authors can analyze whether we can save more compute in the ascent or the descent steps? Probably you can control the training time but vary the ratio of K1/K2. I assume that the ascent step can be performed with fewer example thus more savings.
- For the BERT finetuning result, the authors should include the variance as it's know to be pretty high.
- There are some other efficient SAM approaches that the authors can consider to compare with. For example, only performing SAM every k step[1], etc.

Liu et al. "Towards Efficient and Scalable Sharpness-Aware Minimization." CVPR 2022.

**Summary Of The Paper:**

This paper proposes to sample two subsets from each batch (choosing examples with the largest loss) to perform the SAM algorithm. The authors justify their method by showing the cosine similarity between the gradients obtained before and after the subsample process. Empirical results on CIFAR-10/100, ImageNet, and BERT finetuning show the effectiveness of the proposes method.

**Summary Of The Review:**

This paper has limited novelty, and I've got some questions for the authors (refer to the weaknesses part). Empirical results is not strong enough to overcome the drawbacks. So I vote for rejection for now.

---

> ### Author Response · Authors · 2022-11-17
> **Thanks for the review!**
>
> 1. **How do you obtain the K1 and K2 example specifically? Do you just randomly choose from the K1+K2 examples that have the largest loss?**
>
>     We choose the examples with the largest K1 or K2 loss,  and randomly permute these indices of K1 or K2 examples. We did not observe a difference among the orders of these examples.
>
> 2. **One possible reason is that the batch size 128 used here is pretty small. Can the authors also provide analysis with a larger batch size?**
>
>     Thanks for the suggestion. We actually provide the results in appendix table 10. We will add more analysis. But even with the main batch size (128) we test on, we show that our method is better than random-k which actually drops the accuracy on cifar datasets.
>
> 3. **The reported training time shows that K-SAM can achieve speedup compared to SGD, while on ImageNet it's much slower. Can the authors provide metrics like FLOPs, which is not impacted by the hardware.**
>
>     Thanks for the suggestion. We actually provide the theoretical analysis for the speedup in Section 3.2. But we agree that adding FLOPs will be nicer. The main reason for the slower speed of ImageNet is that we have to choose larger K1 and K2 to make the result comparable to SAM.
>
> 4. **Whether we can save more compute in the ascent or the descent steps?**
>
>     Thanks for the suggestion, we will add the mentioned experiments. Based on our findings, we can save more in the ascent step and that’s why we typically choose K1=B/8 (the ascent step) and K2 = B/2 (The descent step).
>
> 5. **For the BERT finetuning result, the authors should include the variance as it's known to be pretty high.**
>
>     Thanks for the suggestion, we will add them.
>
> 6. **There are some other efficient SAM approaches that the authors can consider to compare with.**
>
>     Thanks for pointing it out, we will add it to the related work.

---

> > ### Comment · Reviewer_Y4Jo · 2022-12-11
> > **Thanks for the reponse**
> >
> > I appreciate the effort in the rebuttal phase. However, my concerns still exist and some questions are not addressed.
> > So I decide to maintain the previous recommendation.

---

### Official Review · Reviewer_wYVX · 2022-10-23

**Confidence:** 5
**Clarity, Quality, Novelty And Reproducibility:** 1. Clarity
**Correctness:** 4
**Technical Novelty And Significance:** 2
**Empirical Novelty And Significance:** 4
**Recommendation:** 6

**Strength And Weaknesses:**

The main weakness of this work is that it lacks novelty and originality. The idea of choosing the top-k elements of the batch with highest loss is natural and has been explored before. In particular the work of Kawaguchi and Lu 2020 present the idea in its most generic form and the presented algorithm is almost a simple application of the generic method in the context of SAM. The only difference being that because SAM is a two-step process, one has to choose a subset of the batch both for the ascent step and the descent step. I wouldn't be surprised if there were even older references before 2020 exploring the exact same idea of Top-k elements in a batch. Authors should be more transparent about the fact that their algorithm is a special case of that presented in Kawaguchi and Lu 2020, applied to SAM (perhaps with some minor modifications).

Another weakness is that the authors overstate the computational cost of SAM compared to SGD. It is known that SAM has double the cost of vanilla SGD, but traditionally a constant factor (of 2 in this case) is not necessarily a reason to worry or discard a method as "too slow". This is reserved for methods whose complexity increases beyond constants. For example SGD which needs constant time to compute an update, vs GD which has linear complexity in the size of the data. In the third paragraph of the introduction statements like "the additional cost may make SAM too expensive for widespread adoption" are simply an exaggeration and should be rewritten.

Even though I started stating the weaknesses, I had a positive impression of the work, mainly due to the extensive experimental evaluation. In my view, the strengths of the paper are twofold:
1. The problem is relevant and significant to some extent: it is great to halve the time required to achieve the benefits of SAM
2. The extensive experimental results are convincing and confirm that it is indeed possible to apply Top-K in the context of SAM without a noticeable degradation of the results. The experiments presented cover many different settings like (a) different architectures (b) different and relevant baselines (c) different tasks like supervised learning with images and NLP tasks (d) different hyperparameter settings and (e) different computational infrastructure (single node vs distributed training). Overall I find the authors did a great job in this regard.

**Summary Of The Paper:**

The authors study the effectiveness of an iterative stochastic minimization algorithm for SAM (sharpness aware minimization): at each iteration the batch is further reduced in size using a Top-K rule based on the value of the loss. The goal of this choice is to reduce the computational expense, taking the cost closer to vanilla SGD while hopefully retaining the benefits of full-batch SAM. Extensive experiments show that the approach is feasible and it is possible to keep almost the same performance of SAM while reducing the computation time to a similar level to that of SGD.

**Summary Of The Review:**

The authors present an application of Ordered SGD (Kawaguchi and Lu 2020) in the context of SAM, showing through extensive and convincing experimental evidence that it allows almost a halving of the training time, while retaining almost the same accuracy numbers from the vanilla SAM method. I don't see a strong reason to reject the paper. The lack of technical novelty might be compensated by the experimental evaluation. It might be useful for practitioners in need of halving the cost of performing SAM.

---

> ### Author Response · Authors · 2022-11-17
> **Thanks for the review!**
>
> 1. **Overstate the computational cost of SAM compared to SGD.**
>
>     Thanks for the suggestion, we may emphasize that SAM is twice as much as vanilla SGD. Although it is doable, it’s much slower compared to SGD. Of course “much slower” is arguably subjective, but considering large-scale training of ML models, we believe SAM would be much more widely adopted in the community, if there was no additional cost during training, compared to a baseline of SGD.
>
> 2. **Make batch subsets in section 3 clear**
>
>     Sure, we will make the point of how we select the subset clear.

---

### Official Review · Reviewer_gf5z · 2022-10-26

**Confidence:** 5
**Correctness:** 2
**Technical Novelty And Significance:** 1
**Empirical Novelty And Significance:** 1
**Recommendation:** 3

**Clarity, Quality, Novelty And Reproducibility:**

* This work directly applies the idea of Ordered-SGD or biased stochastic optimization in the context of sharpness aware minimization.
* There is no concern on reproducibility.


**Strength And Weaknesses:**

Strength
* clarity; the main idea of the paper is described straightforward so K-SAM is read easily.

Weakness
* very limited novelty; this work naively applies the idea of Ordered-SGD to SAM, not more really.
* overclaiming based on shallow analysis; Figure 1 is clearly showing that the gradient estimate based on less number of samples gets more and more dissimilar to full mini-batch gradient, but the authors drive discussion into a direction where this result unfairly gives grounds for their proposed idea. This is an obvious trade-off between approximation accuracy and computational efficiency.
* lack of theoretical grounds; the authors borrow some explanations from Ordered-SGD on the convergence analysis, but without any extension on mini-max as in the SAM setting.
* insufficient empirical evidence; the experiments do not show any significant improvements on both performance or efficiency, except for the expected trade-off.


**Summary Of The Paper:**

This work proposes K-SAM as an alternative to SAM to improve algorithmic efficiency. The basic idea is to reduce the number of training samples used to approximate the stochastic gradients in both inner and outer optimization steps based on their loss values. This idea is originally proposed in Ordered-SGD, a biased stochastic optimization method. The authors argue that the proposed methodology can reduce the amount of computations by choosing K well without losing much on the generalization effect by the original SAM.


**Summary Of The Review:**

Unfortunately this work does not provide non-trivial addition to algorithmic extension to the original SAM or empirical/theoretical evidence to support the realization of the proposed idea.

(after rebuttal) It is my regret that my concerns are not effectively addressed by the authors. I'm still not convinced that the paper is ready for publication.

---

> ### Author Response · Authors · 2022-11-17
> **Thanks for the review.**
>
> 1. **Very limited novelty; this work naively applies the idea of Ordered-SGD to SAM, not more really.**
>
>     We agree that our method builds upon Ordered-SGD. However, applying Ordered-SGD to SAM naively will not achieve a speed as fast as SGD since to achieve this with orders for both directions requires at least two forward pass on the whole mini-batch. Yet, our proposed strategy can achieve such a speed-up. We also show empirically that our method can achieve comparable results to SAM while as fast as SGD on CIFAR datasets that have not been shown before.
>
> 2. **Overclaiming based on shallow analysis**
>
>     We want to highlight that we provide an approximation to SAM when we use our estimation. We show in practice that even without the full training batch, KSAM can achieve comparable accuracy. And Figure 1 shows the intuition of why KSAM may work and why it works better than random k.
>
> 3. **Insufficient empirical evidence**
>
>     Based on our experiments, we show that trade-offs between SAM and KSAM exist and we evaluate the value of KSAM on a series of experiments on several datasets, even for ImageNet classification. We think it is still interesting to see that we can actually get a similar accuracy as SAM when only subsets are applied for both directions.

---

### Author Response · Authors · 2022-11-17
**Thank you all for the reviews!**

Thank you all for the careful review and useful suggestions. We will modify the paper based on the suggestions, and we answer the questions in detail for each reviewer.

---

### Decision · Program_Chairs · 2023-01-20

**Decision:**

Reject

**Justification For Why Not Higher Score:**

The overall ratings are on the reject side and mainly based on the valid criticism that the submission in its current state misses some crucial baselines to compare with.

**Justification For Why Not Lower Score:**

N/A

**Metareview: Summary, Strengths And Weaknesses:**

This paper studies a timely and practically relevant research problem: how to reduce the computational cost of SAM algorithm, which has received a lot of attention recently due to its improved generalization performance. The paper proposes a simple and efficient approach for doing this. It builds on Ordered-SGD approach which selects a subset of K samples with the largest loss value. The authors have observed that such selection criterion can also be applied in the context of SAM (for both descent and ascent steps). This enables SAM to be used on larger scale problems with little performance degradation (although the authors have observed and hypothesize that the level of degradation might be sensitive to the number of classes).

The main concerns raised by reviewers were around novelty and lack of satisfactory baseline comparison. I do not personally believe that a combination of existing components (in this case SAM and Ordered-SGD), which was never tried before, is necessarily a weakness, as long as it comes with strong theoretical and/or empirical justification. The paper is in the empirical category, but reviewers feel that the empirical results in the paper need comparison with a greater set of baselines along the efficiency dimension of SAM. For example, Reviewer Y4Jo suggests including variance for language model results (due to the high variance of performance in these tasks) to ensure the reported numbers are not from a lucky run. In addition, Reviewer Y4Jo and T2UG request additional baselines to be compared against. Specifically, T2UG suggests top-K-SGD with K-SAM such that K=K1+K2 as an absolutely crucial baseline to include for all experiments and contrast the results for K-SAM, and Y4Jo suggests comparing against the work by Liu et al. "Towards Efficient and Scalable Sharpness-Aware Minimization." CVPR 2022.

All in all, I find this work very interesting with a great potential of having practical impact. However, with the current state of the paper and the current reviews, unfortunately the submission cannot be accepted. Yet, I truly do encourage the authors to strengthen their story by considering a broader set of baselines and resubmit their paper.